# Prevalence of SARS-CoV-2 infection in Baja California, Mexico: Findings from a community-based survey in February 2021 in the Mexico-United States border

Oscar E. Zazueta[1ʘ]*, Richard S. Garfein[2‡], J. Oggun Cano-Torres[1‡], César A. Méndez-Lizárraga[1‡], Timothy C. Rodwell[3‡], Raquel Muñiz-Salazar[4‡], Diego F. Ovalle-Marroquín[1‡], Neiba G. Yee[1‡], Idanya Rubí Serafín-Higuera[4‡], Susana González-Reyes[4‡], Jesus Rene Machado-Contreras[4‡], Lucy E. Horton[5‡], Steffanie A. Strathdee[3‡], Ruth Rodríguez[6‡], Linda Hill[3‡], Ietza Bojórquez-Chapela[6ʘ]

1 Department of Epidemiology, Secretariat of Health of Baja California, Mexicali, Baja California, Mexico, 2 Herbert Wertheim School of Public Health, University of California San Diego, San Diego, California, United States of America, 3 Department of Medicine, University of California San Diego (UCSD), San Diego, California, United States of America, 4 Department of Medicine, Universidad Autónoma de Baja California (UABC), Mexicali, Baja California, Mexico, 5 Division of Infectious Diseases and Global Public Health, Department of Medicine, University of California San Diego, La Jolla, California, United States of America, 6 Department of Population Studies, El Colegio de la Frontera Norte (El Colef), Tijuana, Baja California, Mexico

ʘ These authors contributed equally to this work.
‡ RSG, JOC-T, CAM-L, TCR, RM-S, DFO-M, NGY, IRS-H, SG-R, JRM-C, LEH, SAS, RR and LH also contributed equally to this work.
* oez512@mail.harvard.edu

## Abstract

Between March 2020 and February 2021, the state of Baja California, Mexico, which borders the United States, registered 46,118 confirmed cases of COVID-19 with a mortality rate of 238.2 deaths per 100,000 residents. Given limited access to testing, the population prevalence of SARS-CoV-2 infection is unknown. The objective of this study is to estimate the seroprevalence and real time polymerase chain reaction (RT-PCR) prevalence of SARS-CoV-2 infection in the three most populous cities of Baja California prior to scale-up of a national COVID-19 vaccination campaign. Probabilistic three-stage clustered sampling was used to conduct a population-based household survey of residents five years and older in the three cities. RT-PCR testing was performed on nasopharyngeal swabs and SARS-CoV-2 seropositivity was determined by IgG antibody testing using fingerstick blood samples. An interviewer-administered questionnaire assessed participants' knowledge, attitudes, and preventive practices regarding COVID-19. In total, 1,126 individuals (unweighted sample) were surveyed across the three cities. Overall prevalence of SARS-CoV-2 infection by RT-PCR was 7.8% (95% CI 5.5–11.0) and IgG seroprevalence was 21.1% (95% CI 17.4–25.2). There was no association between border crossing in the past 6 months and SARS-CoV-2 prevalence (unadjusted OR 0.40, 95%CI 0.12–1.30). While face mask use and frequent hand washing were common among participants, quarantine or social isolation at home to prevent infection was not. Regarding vaccination willingness, 30.4% (95% CI

**Data Availability Statement:** The survey tool and the dataset were sent for publication to Dryad: https://doi.org/10.5061/dryad.547d7wmbk.

**Funding:** This work was supported by the California Healthcare Foundation, and the funds were managed through the International Community Foundation. The funding institution had no role in the design of the study or collection, analysis, or interpretation of data or manuscript writing.

**Competing interests:** The authors have declared that no competing interests exist.

24.4–3 7.1) of participants said they were very unlikely to get vaccinated. Given the high prevalence of active SARS-CoV-2 infection in Baja California at the end of the first year of the pandemic, combined with its low seroprevalence and the considerable proportion of vaccine hesitancy, this important area along the Mexico-United States border faces major challenges in terms of health literacy and vaccine uptake, which need to be further explored, along with its implications for border restrictions in future epidemics.

## Introduction

SARS-CoV-2 is a novel respiratory coronavirus that was first reported in Wuhan, China, in December 2019. It was declared a Public Health Emergency of International Concern by the World Health Organization (WHO) on January 31, 2020, and later characterized as a pandemic [1]. As of January 26, 2022, more than 360 million COVID-19 cases and 5.6 million deaths had been reported across the world [2]. However, case reporting depends on a myriad of factors, including testing capacities, type of tests used, surveillance system strategy and population health behaviors. Since many of SARS-CoV-2 infections are mild or asymptomatic, they are less likely to be detected by passive surveillance systems. Therefore, SARS-CoV-2 prevalence estimates might be more accurate using population-based studies [3].

Baja California is a state located in the northern part of Mexico that shares a border with California in the United States (U.S.). The border region is demographically and economically important to both countries, particularly at the Tijuana/San Diego and Mexicali/Calexico ports of entry. In 2019 alone, northbound border crossing estimates were 18.5 million pedestrians, 31.3 million personal vehicles and 1.4 million commercial vehicles annually [4]. These activities accounted for nearly 60 billion US dollars in bilateral trade [5]. As part of an international effort to reduce viral transmission in the border region, the U.S. reached agreements with Mexico to limit all non-essential travel across their borders starting March 20, 2020 [6].

The Mexican national surveillance system strategy employed during the COVID-19 pandemic was based on registering only symptomatic individuals who met the working definition of a suspected case and testing 10% of mild cases and 100% of severe acute respiratory illnesses [7]. Therefore, cases with mild or no symptoms went undetected, likely contributing to an increased spread of local SARS-CoV-2 transmission and underreporting of cases. As of January 26, 2022, through the Epidemiologic Surveillance System for Respiratory Diseases (SIS-VER, in Spanish), Baja California has registered 116,870 confirmed cases of COVID-19 [8]. Nationally, this state had the second-highest mortality rate after Mexico City, with 11,451 confirmed deaths (320 deaths per 100,000 population) [9] and the eleventh highest rate of excess mortality (44.9%) due to COVID-19, out of 32 states [10]. Given the passive nature of the Mexican surveillance strategy and the limitations of a sentinel-based approach, population-based COVID-19 prevalence estimates in Baja California were needed to assess the state of the pandemic and inform future health policies going forward as COVID-19 vaccination efforts began. Based on the national policy on COVID-19 vaccine prioritization, during the month of February 2021 only health workers and adults aged 60 or more were to be vaccinated as vaccines arrived in the Baja California [11]. During the study period, only 11,476 residents of Baja California (0.3% of the total population) had received a vaccine for COVID-19 [12]. Our study aim was to estimate the prevalence of SARS-CoV-2 infection using a population-based survey in the three major cities of Baja California prior to implementation of a national vaccination campaign.

## Methods

### Design

We conducted a population-based household survey to estimate the prevalence of SARS-CoV-2 infection by RT-PCR and the IgG seroprevalence in the general population of Baja California, Mexico.

### Participants and study settings

The survey was conducted in Baja California's three most populous cities: Mexicali, Tijuana, and Ensenada from February 1st to February 19th of 2021, immediately following the state's second wave of new cases of COVID-19 (Fig 1). Survey inclusion criteria were: Spanish speakers residing in Baja California for at least six months, age five years or older. Prior to participation, we obtained written informed consent (or assent and parental consent for minors) for biological sample collection and survey interview.

The protocol was approved by the Institutional Review Board (IRB) of Tijuana General Hospital (No. CONBIOETICA-02-CEI-001-20170526). Since data shared with co-authors from San Diego were de-identified, the human subjects review was not required by the University of California, San Diego IRB.

### Sample size and design

A probability, stratified, three-stage clustered sample design was employed. The target sample size (n = 1,500) was calculated to estimate a prevalence of 4% with a 30% precision relative to the expected proportion (1.2% absolute precision), 95% confidence limits and a design effect of 1.9. The sample was stratified by city, and within cities, using the primary sampling unit (PSU) termed "Basic Geo-Statistic Areas" (AGEB), which is the basic unit of Mexico's National

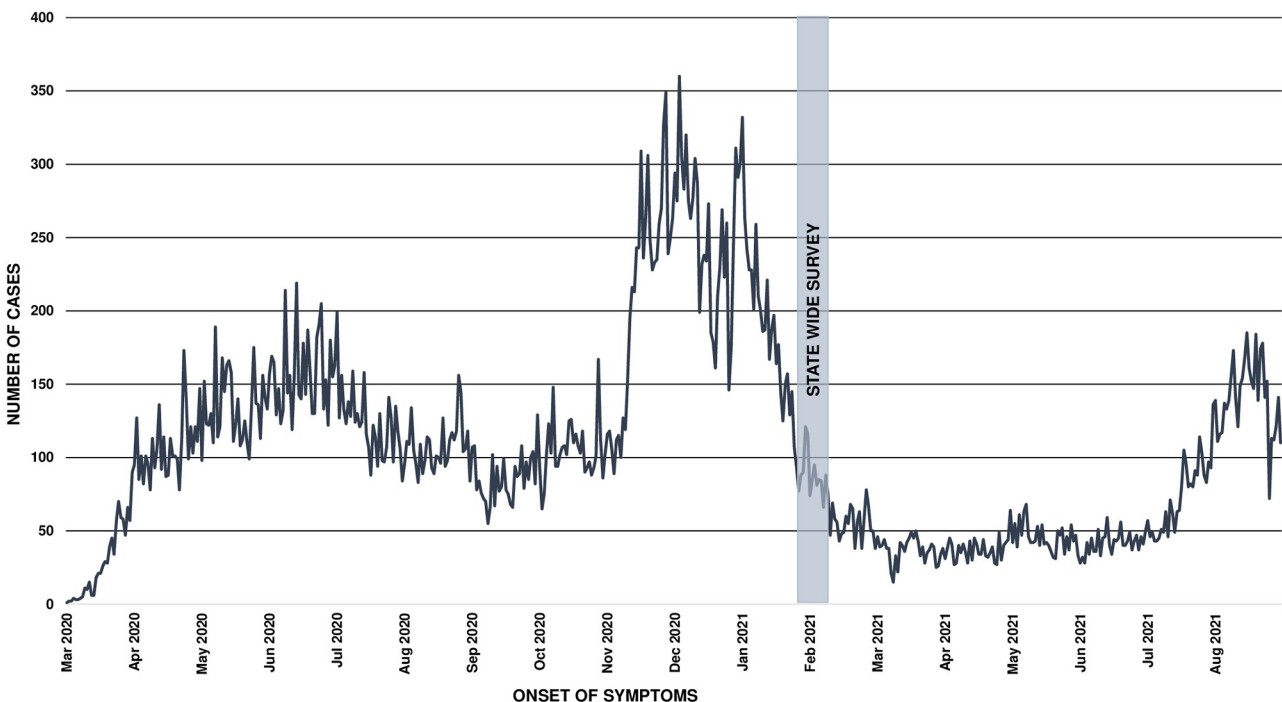

**Fig 1. Epidemic curve in Baja California during the COVID-19 pandemic in 2020–2021.**

Institute of Geography and Statistics (INEGI) for the subdivision of municipal geostatistical areas. Within each stratum, 33 AGEBs were selected with probability proportional to the size (PPS) of the number of inhabitants in the AGEB. Within AGEBs, eight blocks were selected, also with PPS, and in each block four households were selected through systematic random sampling. In the final stage, one participant was selected at random from a list of eligible household members. If the selected household member was not home at the time, the survey teams returned at an appointed time to complete the survey procedures.

## Survey tool and materials

A questionnaire was developed to assess knowledge, attitudes, and practices related to the COVID-19 pandemic, along with sociodemographic data and other variables. Data were captured during a face-to-face interview with each participant using tablets and smartphones equipped with a digital platform for that purpose.

Three biological samples were obtained from each participant: Fingerstick whole blood that was placed onto Whatman 903 protein saver cards for IgG antibody testing of dried blood spots; a nasopharyngeal and oropharyngeal swab for RT-PCR, and a second nasopharyngeal swab for Panbio COVID-19 Ag rapid test device by Abbott (Lake Country, IL, U.S.A.).

The questionnaire and specimen collection were carried out by medical and health sciences students previously trained on the use of personal protection equipment, sampling, and interviewing techniques, with supervision by faculty and physicians from the research team. The questionnaire and the dataset for this survey is published and is open for consultation [13].

## Specimen testing

RT-PCR for all samples was conducted at Baja California Public Health State Laboratory. The extraction of total RNA from oropharyngeal and nasopharyngeal samples used a volume of 200 μL. A 200 μL sample of APEX BioResearch Products Water UltraPure Free of DNAse, RNAse, Proteases and Endonucleases molecular biology grade water was included as negative extraction control. Total RNA was extracted by the Bioneer ExiPrep 96 Viral DNA/RNA kit extraction with magnetic beads, using the Bioneer EP 96L-BXDOO7 automated extraction equipment with a 200 μL sample volume, eluted in 100 μL and stored at 4 ˚C.

RT-PCR was performed targeting the SARS-CoV-2-specific nucleocapsid (N1) gene and human RP gene. Real-time RT-PCR Primers and Probes (2021 Integrated DNA Technologies, Inc.). For N1 and RP, primers and probes came in a single reagent and were used per manufacturer's instructions. Amplification for N1 and RP was carried out separately in a final volume of 20 μL with the following reagents: 10 μL of Master Mix (qPCR BIO Probe 1-Step Go Mix No-ROX), 1μL of 20X Rtase Go Probe qPCR BIO Probe 1-Step Go No-ROX, 1.5 μL of N1 and RP primers/probe premixed reagent, 2.5 μL of PCR grade $H_2O$, and 5 μL of the extracted template. Thermal cycling conditions included 10 min at 50 ˚C for reverse transcription, 2 min at 95 ˚C for polymerase activation, followed by 45 cycles at 95 ˚C for 3 s and 55 ˚C for 30 s for denaturation and amplification/detection, respectively. The RT-PCR was performed on the ABI 7500 real-time PCR (Applied Biosystems, CA, USA). A sample was considered positive if the cycle threshold ($C_t$) for N1 was $\leq 40$ and for RP amplification was $C_t$ of <35 cycles. Samples with RP $C_t$ values >35 were repeated from RNA extraction. If the result was the same, samples were reported as indeterminate. Each sample was evaluated once, positive, negative and non-template controls were included in each experiment.

For antibody detection, whole blood samples were obtained by fingerstick and collected in Whatman 903 protein saver cards. The samples were subsequently sent to the Broad Institute

Serology Lab (BISL, Boston, U.S.), where anti-SARS-CoV-2 IgG presence and abundance were determined in dried blood spots by ELISA assay according to BILS protocol [14].

### Statistical analysis

For prevalence estimates, we employed weights reflecting the inverse of the selection probability, as well as an adjustment for non-response and calibration to match the 2020 Census population of each city. Means and standard deviations were obtained to describe quantitative variables, and absolute and relative frequency for categorical variables. Unadjusted and adjusted prevalence odds ratios (pOR) were calculated to identify sociodemographic, health, and behavioral factors associated with RT-PCR positivity and IgG seroprevalence for SARS-CoV-2 infection using logistic regression analysis. For all tests, p-values <0.05 were considered statistically significant. All statistical analyses were carried out using the complex sample module of Stata STATA 15 (StataCorp, College Station, TX, USA).

## Results

Out of 2,898 households visited, 1,283 (44%) agreed to participate. At the individual level, 1,126 (89%) of the 1,267 randomly selected persons agreed to participate (Fig 2).

### General characteristics

The survey was applied to 1,126 consenting participants, 35% of whom were from Mexicali, 35% from Tijuana, and 30% from Ensenada. After weighting, they represented a population of 2.8 million residents (Fig 2). The mean age was 37 years (95%CI 35–40). Overall, the sample included 50% women, of which 2% reported being currently pregnant (Table 1). Half of the population (53%) had less than a high school education, 26% had completed high school, and 22% had a college degree or higher. Regarding self-perceived health, only 1% of the population

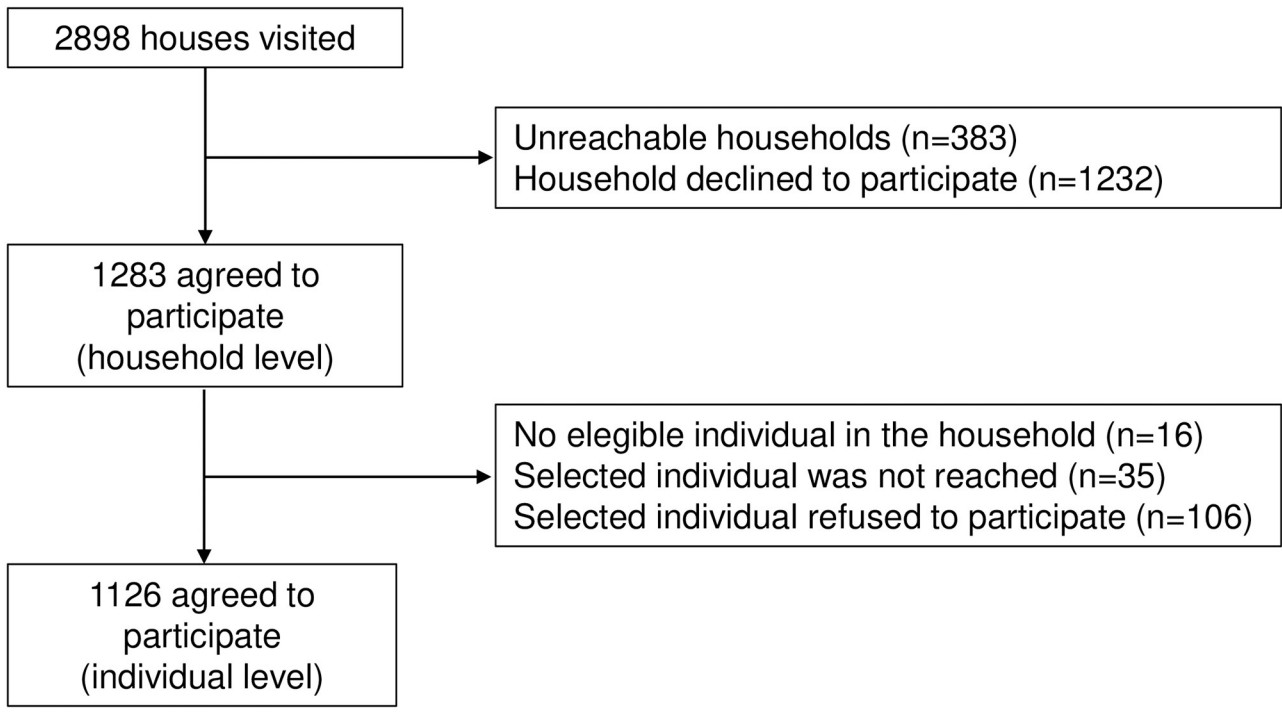

**Fig 2. Flowchart of the study selection process.**

**Table 1. Participant characteristics among a community-based sample of Baja California residents–February, 2021[a].**

| Characteristic | Overall |
|---|---|
| | % or mean (95% CI) |
| Gender (%) | |
| Male | 50 (44,56) |
| Female | 50 (44,56) |
| Age (mean) | 37 (35,40) |
| Age category (years) | |
| 5–17 | 22 (16,30) |
| 18–59 | 67 (60,73) |
| $\geq 60$ | 11 (9,13) |
| Educational level (%) | |
| Elementary or less | 36 (28,43) |
| Secondary | 17 (13,21) |
| Highschool | 26 (21,32) |
| College or more | 22 (17,28) |
| Crossed the border in the last six months (%) | 5 (3,10) |
| Self-reported health status (%) | |
| Very good | 20 (16,25) |
| Good | 56 (50,61) |
| Regular | 23 (19,28) |
| Bad | 1 (1,3) |
| Very bad | 0 (0,1) |
| Received COVID-19 vaccine (%) | 0 (0,2) |
| Currently pregnant (%) | 2 (1,3) |
| Self-reported comorbidities (%) | |
| Diabetes | 12 (10,15) |
| Obesity | 26 (21,31) |
| Hypertension | 21 (17,24) |
| Cardiovascular disease | 6 (5,8) |
| Chronic pulmonary disease | 6 (5,8) |
| Cancer | 2 (1,3) |
| HIV/AIDS | 0 (0,1) |
| Current smoker | 16 (13,20) |

[a]The data presented in this table is weighted for sampling distribution.

reported their health as *bad* or *very bad*, with most participants (76%) reporting *good* or *very good* health status. Since the population-wide vaccination campaign had not started yet in Baja California, less than 1% reported being vaccinated against COVID-19 at the time of the survey. Crossing the border in the past six months was also uncommon (5%). Health risks reported most often were obesity (26%), hypertension (21%), diabetes (12%), and smoking (16%). Sero-positivity was marginally more frequent among females (25%) as compared to males (17%), and no other differences by sociodemographic characteristics were observed (Table 2).

## COVID-19 knowledge, attitudes and practices

Overall, 6% of participants reported not knowing what COVID-19 was, 58% correctly identified it as a disease caused by a virus, and a further 15% knew it was an infectious disease.

**Table 2. Distribution of IgG seropositivity, by participant characteristics, among a community-based sample of Baja California residents–February, 2021[a].**

| Characteristic | IgG negative | IgG positive |
|---|---|---|
|  | % (95% CI) | % (95% CI) |
| Gender |  |  |
| Male | 83 (79,87) | 17 (13,21) |
| Female | 75 (68,80) | 25 (20,32) |
| Age category (years) |  |  |
| 5–17 | 86 (70,94) | 14 (6,31) |
| 18–59 | 78 (74,82) | 22 (18,26) |
| ≥ 60 | 70 (62,77) | 30 (23,38) |
| Educational level |  |  |
| Elementary or less | 84 (77,89) | 16 (11,23) |
| Secondary | 74 (60,84) | 26 (16,40) |
| Highschool | 74 (65,81) | 26 (19,35) |
| College or more | 80 (74,85) | 20 (15,26) |
| Border crossing in the last six months |  |  |
| Did not cross the border | 78 (73,82) | 22 (18,27) |
| Crossed the border | 90 (74,97) | 10 (3,26) |
| Self-reported health status (%) |  |  |
| Very good | 82 (73,89) | 18 (11,27) |
| Good | 79 (73,84) | 21 (16,27) |
| Regular | 78 (67,86) | 22 (14,33) |
| Bad | 47 (17,80) | 53 (20,83) |
| Very bad | 100[b] | 0 |
| Received COVID-19 vaccine |  |  |
| Received vaccine | 79 (26,98) | 21 (2,74) |
| Did not receive vaccine | 79 (74,83) | 21 (17,26) |
| Self-reported comorbidities |  |  |
| Diabetes | 66 (56,75) | 34 (25,44) |
| Obesity | 68 (56,77) | 32 (23,44) |
| Hypertension | 69 (58,79) | 31 (21,42) |
| Cardiovascular disease | 78 (65,87) | 22 (13,35) |
| Chronic pulmonary disease | 75 (59,87) | 25 (13,41) |
| Cancer | 86 (67,95) | 14 (5,34) |
| HIV/AIDS | 0 | 100[b] |
| Current smoker | 93 (86,96) | 7 (4,14) |

[a] The data presented in this table is weighted for sampling distribution.

[b] All cases in this category were IgG negative.

In terms of attitudes on COVID-19, 62% worried about family members or friends getting COVID-19, 49% reported being worried about getting COVID-19 themselves, 12% mentioned being worried about not having access to health services, 14% worried about their job implications if they contracted the disease (such as losing their jobs and finding a new job), 7% worried about not having a safe place to recover from the disease, and 3% worried about being around other people at home.

Regarding prevention measures practiced in the past 6 months to prevent COVID-19 (Table 3), 74% of participants reported hand washing or using hand-sanitizer *very frequently*,

**Table 3. Self-reported COVID-19 prevention measures and risks in the past six months.**

| | Very frequently | Frequently | Occasionally | Rarely | Never |
|---|---|---|---|---|---|
| Washed hands regularly or used hand-sanitizer (%) | 74 | 18.3 | 6.3 | 1.2 | 0.2 |
| Quarantined or socially isolated at home or a shelter (stayed at home/shelter without any visitors from outside the household) | 42.3 | 20.8 | 13 | 9 | 14.9 |
| Avoided leaving the house | 56.3 | 21 | 13.9 | 5.7 | 3.1 |
| Used a facemask or face shield when in public spaces | 85.7 | 12.1 | 0.9 | 1.3 | 0 |
| Used public transportation | 9.9 | 2.8 | 9.5 | 19.9 | 57.9 |
| Visited a park, bar, or restaurant. Went to a concert or other crowded place | 5.7 | 2.3 | 8.1 | 24.8 | 59.1 |
| Visited family members in other households | 6.2 | 8.3 | 20.7 | 26.1 | 38.7 |

86% reported using face mask *very frequently*, 42% reported adopting quarantine or social isolation at home or shelter *very frequently*, and 56% avoided leaving their house *very frequently*.

When asked which measures participants believed might help to prevent COVID-19, the most frequently mentioned responses were wearing a face mask (89%), washing hands with soap and water (66%), and keeping a distance of 1.5 meters from other people (57%).

When asked if they would get vaccinated if they had free access to the vaccine against COVID-19, less than half (45%) reported that it was *very likely* that they would and 30% reported that it was *very unlikely*. When asked about their main concerns about the COVID-19 vaccine, 37% had no concerns, whereas 41% worried about side effects, 10% did not think that the vaccine worked, 4% worried that the development process of the vaccines was too fast, 4% did not trust healthcare providers, 4% worried that health authorities had economic rather than public health reasons to promote vaccines and 1% had concerns about asking permission in their jobs.

### Prevalence and factors associated with SARS-CoV-2 infection

The overall prevalence of SARS-CoV-2 infection detected by RT-PCR was 7.8% (95% CI 5.5–11.0) (Table 4), the highest documented in Ensenada (22.2%, 95% CI 15.0–31.6), followed by Tijuana (6.4%, 95% CI 3.4–11.6), and Mexicali (5.5%, 95% CI 2.9–10.3). The prevalence of IgG antibody seropositivity was 26% (95% CI 20.6–32.3) in Mexicali, 18.7% in Tijuana (95% CI 13.7–25.0), and 21.9% in Ensenada (95% CI 16.2–28.9), with an overall prevalence of 21.1% across the three cities (95% CI 17.4–25.2). We also examined the prevalence of SARS-CoV-2 infection by considering positivity for either RT-PCR or IgG antibodies and observed a weighted prevalence of 28.4% (95% CI 22.5–35.2) in Mexicali, 23.3% (95% CI 17.6–30.2) in Tijuana, 38.6% (95% CI 31.0–46.8) in Ensenada, and 26.3% (95% CI 22.2–30.9) overall.

After adjusting for potential confounders, higher odds of being positive for IgG was associated with being a current smoker (pOR = 3.0, 95% CI 1.5–6.1), to be part of the older age

**Table 4. Prevalence of SARS-CoV-2 infection.**

| LOCATION | Number of participants (n) | RT-PCR | | IgG antibodies | | RT-PCR and IgG antibodies combined | |
|---|---|---|---|---|---|---|---|
| | | Weighted prevalence (%) | 95% CI | Weighted prevalence (%) | 95% CI | Weighted prevalence (%) | 95% CI |
| Mexicali | 413 | 5.5 | 2.9, 10.3 | 26.0 | 20.6, 32.3 | 28.4 | 22.5, 35.2 |
| Tijuana | 373 | 6.4 | 3.4, 11.6 | 18.7 | 13.7, 25.0 | 23.3 | 17.6, 30.2 |
| Ensenada | 340 | 22.2 | 15.0, 31.6 | 21.9 | 16.2, 28.9 | 38.6 | 31.0, 46.8 |
| **Overall** | **1,126** | **7.8** | **5.5, 11.0** | **21.1** | **17.4, 25.2** | **26.3** | **22.2, 30.9** |

**Table 5. Association between COVID-19 (IgG positivity) and exposures.**

| Exposure | pOR | 95% CI | Adjusted pOR[a] | 95% CI |
|---|---|---|---|---|
| Gender (Ref.: male) | 1.7 | 1.1,2.6 | 1.4 | 0.9,2.2 |
| >60 years old | 1.7 | 1.1,2.7 | 1.8 | **1.0,3.2** |
| Educational level | | | | |
| Elementary or less (ref.) | | | | |
| Secondary | 1.8 | 0.9,3.8 | 1.7 | 0.9,3.4 |
| High school | 1.8 | 1.0,3.3 | 2.2 | **1.2,4.1** |
| College or more | 1.3 | 0.8,2.2 | 1.6 | 0.9,2.8 |
| Worked during past week | 1.2 | 0.7,1.9 | 1.0 | 0.6,1.6 |
| Crossed the border in the past 6 months | 0.4 | 0.1,1.3 | 0.4 | 0.1,1.4 |
| History of diabetes | 0.5 | 0.3,0.8 | 0.6 | 0.3,1.1 |
| History of obesity | **0.4** | **0.3,0.8** | 0.6 | 0.1,1.0 |
| History of hypertension | 0.5 | 0.3,0.9 | 0.8 | 0.4,1.5 |
| History of cardiovascular disease | 0.9 | 0.5,1.7 | 1.3 | 0.6,2.7 |
| History of chronic pulmonary disease | 0.9 | 0.4,1.9 | 1.1 | 0.5,2.4 |
| History of cáncer | 1.4 | 0.5,3.7 | 2.3 | 0.9,5.8 |
| Smoking | **3.8** | **1.9,7.5** | **3.0** | **1.5,6.1** |

[a]pOR were adjusted for all variables included in the table.

group (pOR = 1.8, 95% CI 1.0–3.2), and to have high school education as compared to elementary or less (pOR = 2.2, 95% CI 1.2–4.1) (Table 5).

## Discussion

This study shows that in February of 2021 the population-based weighted estimate of SARS-CoV-2 infection by RT-PCR in the three largest cities of Baja California was high (7.8%), with a higher prevalence in Ensenada as compared with Mexicali and Tijuana. Overall, SARS-CoV-2 IgG weighted seroprevalence was 21.1% and was similar across the three cities. Active smoking was associated with higher odds of infection based on serum IgG antibodies, while border crossing in the past 6 months, working during the last week, and history of diabetes, obesity, hypertension, cardiovascular disease, chronic pulmonary disease, or cancer were not associated with COVID-19 seroprevalence. COVID-19 knowledge in the general population was limited overall; however, risk reduction practices like wearing face masks, washing hands regularly, and social distancing were common, while few participants reported home isolation or quarantine and avoiding indoor locations to prevent infection. Hesitancy surrounding vaccine uptake before the Mexico national vaccination campaign was reported by more than half the participants, with approximately one-third reporting that they were unsure or unlikely to get vaccinated even if it was available free through public health services.

In this study, Ensenada had a considerable higher prevalence of SARS-CoV-2 infection by RT-PCR at the time of the survey as compared to Tijuana and Mexicali. This finding is consistent with official data from the National Epidemiological Surveillance System [8], and it is a reflection of the dynamics of the epidemic over time within the same territory. This differences could be attributed to the fact that Mexicali and Tijuana were experiencing the decrease phase of the second wave of COVID-19, while Ensenada had a slower decrease rate when the survey took place. This statement is supported by official data for the period of January 1st–February 19th of 2021, when the incidence rate in Ensenada (539 cases per 100,000 habitants) was 2.8 times higher than in Tijuana (189.3 cases per 100,000 habitants) and 2.1 times higher than in

Mexicali (251.7 cases per 100,000 habitants) [8, 15]. Also, it is important to note that these incidence rates have to be interpreted carefully, since they come from a sentinel surveillance system. Furthermore, mortality data due COVID-19 also seems to support these conclusions. For this time period (January 1st–February 19th, 2021), Ensenada had a much higher mortality rate (107.9 deaths per 100,000 habitants) compared to Tijuana and Mexicali (39.7 and 50.2 deaths per 100,000 habitants, respectively) [8, 15].

Seroprevalence results obtained in this study were slightly lower than those reported by the National Survey on Health and Nutrition (ENSANUT in Spanish), conducted in Mexico in August-November 2020 [16]. However, it should be noted that while the nationwide seroprevalence of anti-SARS-CoV-2 IgG antibodies was 24.9% (95%CI 22.2–26.7%), the Pacific-Northern region, which includes Baja California and four other border states, had a seroprevalence of 31% (95%CI 25–36.6%) from a smaller sample (n = 851).

An additional cross-sectional study conducted in 34 clinical laboratories and 34 blood banks from the Mexican Institute of Social Security (IMSS, in Spanish) found that the overall seroprevalence based on IgG antibodies in Mexico was 3.5% in February of 2020 but increased to 33.5% by December 2020 [17]. They estimated a seroprevalence of 40.7% (95% CI 36.9–44.5%) in the Northwest region, which is significantly higher than what our study documented. The study analyzed 24,273 serum samples from across all 32 States; however, the observed differences may be explained by the study design and potential selection bias associated to including only samples from blood banks and clinical laboratories.

Conversely, the seroprevalence in Baja California was much higher than in other regions of Latin America, including Ecuador (n = 2,457, seroprevalence: 13.2%) [18], Brazil (n = 31,128, seroprevalence: 2.8%) [19], Chile (n = 1,367, seroprevalence: 13.3%) [20], and Argentina (n = 2,024, seroprevalence: 10.1%) [21]. Nonetheless, one study in Colombia conducted from September-October 2020 showed a higher prevalence in the three samples cities; Medellín (n = 1,832, seroprevalence: 27%), Barranquilla (n = 1,487, Seroprevalence: 55%), and Leticia (n = 1,417, seroprevalence: 59%) [22].

In this study, active smoking was significantly associated with the presence of anti-COVID-19 IgG antibodies in participants, suggesting it was a risk factor for natural infection. This is consistent with studies showing a higher prevalence of neutralizing SARS-CoV-2 antibodies reported among current smokers [23, 24]. However, lower blood levels of anti-SARS-CoV-2 IgG have also been documented in active smokers compared to non-smokers [25], and there is evidence that smoking can decrease serum levels of IgG [26]; indicating that the relationship between SARS-CoV-2 infection and cigarette smoking is complex and likely highly dependent on time since infection.

This study had several limitations. First, a high refusal rate may have impacted the representativeness of the sample. The possibility of selection bias as a result of this was partially addressed by weighting of the results, but still we cannot discard this possibility. Second, there were concerns about the sensitivity of the IgG assay based on dried blood spots instead of venous blood which could have resulted in an underestimate of the seroprevalence in these populations. Third, our study was limited to large urban areas, which may not adequately reflect the prevalence in semi-urban and rural areas within the state. However, we are confident that this effect was minimal given that the population living outside of the three sampled cities represents less than 5% of Baja California's population.

The results of this study have significant policy implications, given that border crossing restrictions were implemented in March of 2020, effectively limiting the entrance of non-citizens to the U.S. [6]. These restrictions were suspended on November 8, 2021, allowing fully vaccinated non-citizen travelers to enter the United States [27]. This study showed that regardless of being a border state, Baja California had a COVID-19 prevalence that was comparable

to the rest of Mexico, and active border crossing in the past 6 months was not associated with a higher prevalence of SARS-CoV-2 infection. Population-based prevalence studies on the other side of the border, particularly in the counties of San Diego and Imperial in California, U.S., would be very valuable to compare the impact of the border restrictions during the pandemic, and would provide more information to assess the effectiveness of this strategy to limit the dynamics of infectious diseases with pandemic potential.

In terms of practice, this study showed that COVID-19 knowledge in the general population was low after the second peak of cases in Baja California, despite the vast amount of information available on the disease. While the use of facemasks and hand washing was frequent, adopting quarantine or home isolation as a risk reduction measure was less common, potentially due to lack of information or the need to conduct essential activities outside the home. Most importantly, 35% of participants reported that they were "unlikely" or "very unlikely" to get the COVID-19 vaccine for a variety of reasons. Policymakers should be aware that over 50% of the participants interviewed expressed concerns about the potential side effects and effectiveness of the vaccine, indicating more work to be done in this population for optimal vaccine uptake in the future.

In conclusion, this study showed that the seroprevalence of SARS-CoV-2 infection in Baja California in February 2021 was comparable to other states in Mexico and that being a border state with the U.S. did not seem to be associated with higher infection rates. The observed higher prevalence of SARS-CoV-2 infection by RT-PCR in Ensenada as compared to Mexicali and Tijuana emphasizes the dynamic nature of the epidemic within the same territory, and highlights the importance of repeated cross-sectional studies over time to capture these differences as the pandemic continues. Finally, health authorities face important challenges in terms of health literacy concerning prevention measures and future vaccination campaigns, as shown by the high level of vaccine hesitancy in this study. Additionally, decision-makers should address possible barriers regarding isolation/quarantine as part of a test and isolate strategy, as this remains an essential public health measure for control of communicable diseases. Additional incentives such as financial support during home isolation may be necessary to encourage compliance with isolation recommendations.

## Acknowledgments

We want to thank the support of the International Community Foundation and the California Healthcare Foundation. The General Consulate of Mexico in San Diego, Carlos González, Manuel Sánchez-Alavez from UABC, and Verónica Bejarano from the State Laboratory of Public Health.

## Author Contributions

**Conceptualization:** Oscar E. Zazueta, Richard S. Garfein, J. Oggun Cano-Torres, César A. Méndez-Lizárraga, Timothy C. Rodwell, Jesus Rene Machado-Contreras, Linda Hill, Ietza Bojórquez-Chapela.

**Data curation:** Ruth Rodríguez, Ietza Bojórquez-Chapela.

**Formal analysis:** Oscar E. Zazueta, Ruth Rodríguez, Ietza Bojórquez-Chapela.

**Funding acquisition:** Linda Hill.

**Investigation:** Oscar E. Zazueta, César A. Méndez-Lizárraga, Jesus Rene Machado-Contreras, Linda Hill, Ietza Bojórquez-Chapela.

**Methodology:** Oscar E. Zazueta, Richard S. Garfein, César A. Méndez-Lizárraga, Timothy C. Rodwell, Diego F. Ovalle-Marroquín, Jesus Rene Machado-Contreras, Ruth Rodríguez, Linda Hill, Ietza Bojórquez-Chapela.

**Project administration:** Oscar E. Zazueta, J. Oggun Cano-Torres, César A. Méndez-Lizárraga, Idanya Rubí Serafín-Higuera, Susana González-Reyes, Jesus Rene Machado-Contreras, Ruth Rodríguez, Linda Hill, Ietza Bojórquez-Chapela.

**Software:** Ruth Rodríguez, Ietza Bojórquez-Chapela.

**Supervision:** Oscar E. Zazueta, Richard S. Garfein, J. Oggun Cano-Torres, César A. Méndez-Lizárraga, Timothy C. Rodwell, Raquel Muñiz-Salazar, Diego F. Ovalle-Marroquín, Neiba G. Yee, Idanya Rubí Serafín-Higuera, Susana González-Reyes, Jesus Rene Machado-Contreras, Lucy E. Horton, Steffanie A. Strathdee, Linda Hill, Ietza Bojórquez-Chapela.

**Validation:** Oscar E. Zazueta, Ietza Bojórquez-Chapela.

**Writing – original draft:** Oscar E. Zazueta, César A. Méndez-Lizárraga.

**Writing – review & editing:** Oscar E. Zazueta, Richard S. Garfein, J. Oggun Cano-Torres, César A. Méndez-Lizárraga, Timothy C. Rodwell, Raquel Muñiz-Salazar, Diego F. Ovalle-Marroquín, Neiba G. Yee, Idanya Rubí Serafín-Higuera, Susana González-Reyes, Jesus Rene Machado-Contreras, Lucy E. Horton, Steffanie A. Strathdee, Linda Hill, Ietza Bojórquez-Chapela.

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
