## [Decision Letter · Decision Letter 0]

13 Apr 2022

PGPH-D-22-00365

Prevalence of SARS-CoV-2 infection in Baja California, Mexico: Findings from a community-based survey in February 2021

Dear Dr. Zazueta,

Thank you for submitting your manuscript to PLOS Global Public Health. After careful consideration, we feel that it has merit but does not fully meet PLOS Global Public Health’s publication criteria as it currently stands. Therefore, we invite you to submit a revised version of the manuscript that addresses the points raised during the review process.

We look forward to receiving your revised manuscript.

Kind regards,

Megan Coffee, MD, PhD

Academic Editor

Journal Requirements:

1. Your co-authors, J. Oggun Cano-Torres (oggunc@gmail.com), César A Méndez-Lizárraga (melc.cesar@gmail.com), Timothy C Rodwell (trodwell@health.ucsd.edu), Neiba G. Yee (neiba.yeemz@gmail.com), Idanya Rubí Serafín-Higuera (serafin.idanya@uabc.edu.mx), Susana González-Reyes (susana.gonzalez.reyes@uabc.edu.mx), Jesus Rene Machado-Contreras (rene.machado@uabc.edu.mx), Lucy E Horton (lhorton@health.ucsd.edu), Steffanie A Strathdee (sstrathdee@health.ucsd.edu), Ruth Rodríguez (beatriz@colef.mx), and Ietza Bojórquez-Chapela (ietzabch@colef.mx), have not confirmed authorship of the manuscript. We have resent them the authorship confirmation email; however please check that the above email address for them is correct and follow up personally to ensure they confirm. Please note that we cannot pass your manuscript to Production until we have received confirmations from all co-authors.

Just in case your co-authors are having difficulty confirming their authorship, you may advise them to send us an email at globalpubhealth@plos.org and we will confirm their authorship on the authors' behalf.

2. Please update your Competing Interests statement. If you have no competing interests to declare, please state: “The authors have declared that no competing interests exist.”

3. In the online submission form, you indicated that your data will be submitted to the Dryad database upon acceptance. Should your submission be accepted, we will require the following information in your Data Availability Statement: 

1. The DOI provided by Dryad

2. The citation for your data package in the reference section of your manuscript

3. The citation for your data package in the methods section

If you are unable to adhere to our open data policy, please kindly revise your statement to explain your reasoning and we will seek the editor's input on an exemption. Please be assured that, once you have provided your new statement, the assessment of your exemption will not hold up the peer review process."

4. We noticed that you used “data not shown” in the manuscript. We do not allow these references, as the PLOS data access policy requires that all data be either published with the manuscript or made available in a publicly accessible database. Please either remove these references, or amend the supplementary material to include the referenced data.

5. We do not publish any copyright or trademark symbols that usually accompany proprietary names, eg (R), (C), or TM  (e.g. next to drug or reagent names). Therefore please remove all instances of trademark/copyright symbols throughout the text, including KoboToolBox® on page 6.

Additional Editor Comments (if provided):

Seroprevalence surveys provide an important snapshot during the COVID pandemic. The pandemic moves quickly so prevalence data and vaccination rates change quickly. I would in particular separate out the information on vaccine. Perhaps remove vaccination status from table 1 and have a separate section discussing that in light of roll out and policy stage at time vaccination rates were low and 'pre-contemplative'. I might provide more info on movement and travel restrictions as well as vaccine roll out at the time, as the early phase of COVID may not be remembered in the future when the paper is read later on.

Reviewers' comments:

Reviewer's Responses to Questions

**Comments to the Author**

1. Does this manuscript meet PLOS Global Public Health’s publication criteria? Is the manuscript technically sound, and do the data support the conclusions? The manuscript must describe methodologically and ethically rigorous research with conclusions that are appropriately drawn based on the data presented.

Reviewer #1: Partly

Reviewer #2: Partly

2. Has the statistical analysis been performed appropriately and rigorously?

Reviewer #1: No

Reviewer #2: Yes

3. Have the authors made all data underlying the findings in their manuscript fully available (please refer to the Data Availability Statement at the start of the manuscript PDF file)?

Reviewer #1: No

Reviewer #2: Yes

4. Is the manuscript presented in an intelligible fashion and written in standard English?

Reviewer #1: Yes

Reviewer #2: Yes

5. Review Comments to the Author

Reviewer #1: The authors carried out a cross-sectional study to evaluate COVID-19 prevalence the state of Baja California, Mexico. The paper is interesting, but there are some pieces of information that should be added to help the interpretation of results. Some specific comments follow:

Lines 49-41: the conclusions of the abstract seem quite different from those of the paper, and not fully supported by the results. The authors should align them.

Line 95: how long was the survey period? It seems unlikely that all subjects were tested in only one day (1-19-2021)

Lines 185-186. It is difficult for the reader to interpret the reported prevalence of vaccinated subjects (<1%) without putting this results in the context of the vaccination policy of Mexico. Some basic information about it should be provided in the introduction section. As national campaigns of immunization started in many countries at the beginning of 2021, the observed result could be not necessarily worrying.

Table 1. row percentages would have been much more informative than column ones. If the Authors report, for example, that seroprevalence was 10% in men and 15% in women, this is something that can be directly interpreted. On the contrary, the reported percentage do not have a clear interpretation. Moreover, the sum of these percentages sometimes is more than 100% (probably due to rounding issues). Please double-check them.

Line 227. One of the most striking results is the much higher RT-PCR prevalence of SARS-CoV-2 in Ensenada (22.2%) compared to Tijuana (6.4%) and Mexicali (5.5). Moreover, it is notable that seroprevalence in Ensenada was 21.9, lower than RT-PCR prevalence, which seems odd, and very similar to the other two cities. How do the authors explain these apparently contrasting results? Were the official figures (cases and deaths) very different among these cities at the beginning of 2021?

Table 4. the estimated effect of vaccination is based on a very small number of subjects (<1% of them were vaccinated) and is thus probably unreliable. For this reason, this variable should not be included in the model.

Lines 278-279. “however, the observed differences may be explained by the study design and potential selection bias.” Some more information is required here. Is the previous or the present study that the Authors are suggesting that was affected by selection bias?

Lines 295-296 “This study had several limitations. First, a high refusal rate may have impacted the representativeness of the sample.” Again, some more information and discussion are required here, as this is a major weakness of your study. The refusal rate seems to be about 60%, thus the possibility that selection bias affected your estimates is pretty high. Is there anything the Authors can say to increase confidence in these results?

Lines 306-309 I assume that the main reason for border crossing restrictions was that prevalence of COVID-19 was much higher on one side than the other, not that active border crossing was itself considered a risk factor for COVID-19 or that Baja California had a higher prevalence than the rest of Mexico. The authors should clarify this point.

Lines 312-313 “While the use of facemasks and hand washing was frequent, adopting quarantine or home isolation as a risk reduction measure was less common…”. While facemask use and hand washing are general rules for the population, quarantine or home isolation apply in specific circumstances (e.g., being tested positive). Thus, it is not completely clear whether it is correct to compare the prevalence of these two types of measures.

Data Availability

The data should be provided as part of the manuscript or its supporting information or deposited to a public repository duing the submission, not after.

Reviewer #2: This is a short-term survey of SARS-CoV2 active infection and seroprevalence in a population in CA Mexico. While the work is of a value to the local community and the country, there are some concerns with the data analyses that can be improved. Here are my comments:

1. First major challenge is that the study was conducted over a short period of time and with one time assessment of the population. A major limitation and should be justified.

2. Second major major challenge is that there are no data provided to distinguish whether seroprevalence and/or active infection is due to infection or vaccination. The authors need to include vaccination data (from the surveys on the subject) and relate them to the infection and seroprevalence status.

3. Minor comments

a. why infection PCR prevalence highest in Ensenada compared to other cities?

b. Table 3. Please add no. subjects for each location.

c. You should conduct stratification analysis by age (and co-morbidities) and smoking status in relation to PCR prevalence and seroprevalence?

6. PLOS authors have the option to publish the peer review history of their article (what does this mean?). If published, this will include your full peer review and any attached files.

**Do you want your identity to be public for this peer review?** For information about this choice, including consent withdrawal, please see our Privacy Policy.

Reviewer #1: No

Reviewer #2: **Yes: **Walid Alali

---

## [Decision Letter · Decision Letter 1]

30 Jun 2022

Prevalence of SARS-CoV-2 infection in Baja California, Mexico: Findings from a community-based survey in February 2021 in the Mexico-United States border

PGPH-D-22-00365R1

Dear Dr Zazueta

We are pleased to inform you that your manuscript 'Prevalence of SARS-CoV-2 infection in Baja California, Mexico: Findings from a community-based survey in February 2021 in the Mexico-United States border' has been provisionally accepted for publication in PLOS Global Public Health.

Thank you for your work. It is important to have this insight into the pandemic.

Best regards,

Megan Coffee, MD, PhD

Academic Editor

Reviewer Comments (if any, and for reference):

Reviewer's Responses to Questions

**Comments to the Author**

1. If the authors have adequately addressed your comments raised in a previous round of review and you feel that this manuscript is now acceptable for publication, you may indicate that here to bypass the “Comments to the Author” section, enter your conflict of interest statement in the “Confidential to Editor” section, and submit your "Accept" recommendation.

Reviewer #1: All comments have been addressed

Reviewer #2: All comments have been addressed

2. Does this manuscript meet PLOS Global Public Health’s publication criteria? Is the manuscript technically sound, and do the data support the conclusions? The manuscript must describe methodologically and ethically rigorous research with conclusions that are appropriately drawn based on the data presented.

Reviewer #1: Yes

Reviewer #2: Yes

3. Has the statistical analysis been performed appropriately and rigorously?

Reviewer #1: Yes

Reviewer #2: Yes

4. Have the authors made all data underlying the findings in their manuscript fully available (please refer to the Data Availability Statement at the start of the manuscript PDF file)?

Reviewer #1: Yes

Reviewer #2: Yes

5. Is the manuscript presented in an intelligible fashion and written in standard English?

Reviewer #1: Yes

Reviewer #2: Yes

6. Review Comments to the Author

Reviewer #1: (No Response)

Reviewer #2: No more comments

7. PLOS authors have the option to publish the peer review history of their article (what does this mean?). If published, this will include your full peer review and any attached files.

**Do you want your identity to be public for this peer review?** For information about this choice, including consent withdrawal, please see our Privacy Policy.

Reviewer #1: **Yes: **Francesco Barone-Adesi

Reviewer #2: **Yes: **Walid Alali
